# Identification and Evolutionary Analysis of Cotton (*Gossypium hirsutum*) *WOX* Family Genes and Their Potential Function in Somatic Embryogenesis

**DOI:** 10.3390/ijms241311077

**Published:** 2023-07-04

**Authors:** Ruibin Sun, Xue Zhang, Dan Ma, Chuanliang Liu

**Affiliations:** 1National Key Laboratory of Cotton Bio-Breeding and Integrated Utilization, Institute of Cotton Research, Chinese Academy of Agricultural Sciences, Anyang 455000, China; sunruibin@caas.cn (R.S.); zhangxue05@caas.cn (X.Z.); madan@caas.cn (D.M.); 2Zhengzhou Research Base, State Key Laboratory of Cotton Biology, Zhengzhou University, Zhengzhou 450001, China

**Keywords:** cotton, somatic embryogenesis, *WOX*, expression, genome wide

## Abstract

WUSCHEL-related homeobox (WOX) proteins participate profoundly in plant development and stress responses. As the difficulty of somatic embryogenesis severely constrains cotton genetic modification, in this study, we identified and comprehensively analyzed *WOX* genes in cotton. As a result, 40 *WOX* genes were identified in the upland cotton genome. All these cotton *WOX* genes were classified into three clades, ancient, intermediate, and modern clades, based on the phylogenetic analysis of previous studies. The majority (24) of the cotton *WOX* genes belonged to the modern clade, in which all gene members contain the vital functional domain WUS-box, which is necessary for plant stem cell regulation and maintenance. Collinearity analysis indicated that the *WOX* gene family in cotton expanded to some degree compared to Arabidopsis, especially in the modern clade. Genome duplication and segmental duplication may greatly contribute to expansion. Hormone-response- and abiotic-stress-response-related cis-acting regulatory elements were widely distributed in the promoter regions of cotton *WOX* genes, suggesting that the corresponding functions of stress responses and the participation of development processes were involved in hormone responses. By RNA sequencing, we profiled the expression patterns of cotton *WOX* genes in somatic embryogenesis. Only about half of cotton *WOX* genes were actively expressed during somatic embryogenesis; different cotton *WOX* genes may function in different development stages. The most representative, *GhWOX4* and *GhWOX13*, may function in almost all stages of somatic embryogenesis; *GhWOX2* and *GhWOX9* function in the late stages of embryo patterning and embryo development during cotton somatic embryogenesis. Co-expression analysis showed that the cotton *WOXs* co-expressed with genes involved in extensive genetic information processing, including DNA replication, DNA repair, homologous recombination, RNA transport, protein processing, and several signaling and metabolism pathways, in which plant hormones signal transduction, MAPK signaling pathways, phosphatidylinositol signaling systems, and ABC transporters, as well as the metabolism of fatty acid; valine, leucine, and isoleucine biosynthesis; and cutin, suberine, and wax biosynthesis, were most significantly enriched. Taken together, the present study provides useful information and new insights into the functions of cotton *WOX* genes during somatic embryogenesis. The specific regulatory roles of some *WOX* genes in somatic embryogenesis are worthy of further functional research.

## 1. Introduction

Genomic modification is widely applied to improve the yield, stress tolerance, disease resistance, and other desired traits of crops. In cotton, Agrobacterium-mediated genetic transformation is the most widely used approach to incorporate exogenous genes into plants. However, the inability of regeneration via somatic embryogenesis from the transformed explant tissues severely constrains cotton genetic engineering. To date, most cotton cultivars with elite traits are difficult to regenerate, and the current established regeneration system in some cultivars is inefficient and time-consuming [1,2]. Though the optimization of cultural conditions improved regeneration to some degree, the regeneration ability is genotype-dependent [3]. The genetic improvement of the regeneration ability is a critical strategy used to overcome the difficulty of genetic transformation in cotton [2]. The identification of regeneration-related functional genes and the clarification of the underlying mechanisms of somatic embryogenesis is important.

The WUSCHEL-related homeobox (WOX) transcription factor is a plant-specific transcription factor family which plays a critical role in the maintenance of undifferentiated pluripotent stem cells, embryonic development, the development of lateral organs, and some other key developmental processes in plants [4,5,6]. WOXs are broadly distributed in plants; in this study, all WOX members were classified into three clades: the ancient clade, intermediate clade, and modern /WUS clade. The ancient clade members exist in all plant species, from algae to higher plants; the intermediate clade members exist only in vascular plant species; and the modern clade members are only found in seed plants [7].

WOXs are characterized by the presence of the DNA-binding homeodomain (HD). The HD in the N-terminal of WOX is conserved in plants, while other regions of the WOX are highly divergent in sequence. The C-terminal region of WOX may contain two functional domains, the WUS-box (TLXLFP, where X can be any amino acid) and the EAR-like motif (SLELRLN) [8]. Functional identification showed that the WUS-box and EAR-like motif in AtWUS are repression effect domains for the regulation of stem cell identity and the size of the meristem, especially for the WUS-box, which is necessary for the activities of the AtWUS [9].

*WUS* expression is required to specify stem cell identity to maintain an appropriate number of pluripotent stem cells in shoot apical meristem, resulting in the regulation of the size of the shoot apical meristem [10,11]. *WUS-1* mutant *Arabidopsis thaliana* plants fail to organize a normal shoot apical meristem, while the ectopic expression of the *WUS* gene induces increased shoot meristems and the formation of adventitious shoots and somatic embryos [12]. In embryogenesis, *WUS* genes regulate vegetative-to-embryonic transition and help maintain embryonic identity [13].

*WUS* as well as some other *WOX* genes have been reported to promote somatic embryogenesis in many plant species, such as tobacco [14,15], wheat [16], maize [17,18], *Picea abies* [19,20], banana [21], and upland cotton [22]. Particularly, co-overexpression of *ZmWUS2* and *BBM* genes significantly promotes the somatic embryogenesis of maize and largely shortens the genetic transformation period [17]. In wheat, an overexpression of wheat *tawox5* dramatically increases wheat transformation efficiency and overcomes the genotype dependency [16]. The *WUS* gene was also identified as participating in somatic embryogenesis in cotton. The ectopic expression of the *AtWUS* gene in upland cotton promotes the formation of embryogenic callus and the somatic embryogenesis of upland cotton CRI24 cultivar, but this did not work in reluctant cultivars [23]. The ectopic expression of *GhWUS* in Arabidopsis induced somatic embryo and shoot formation from an Arabidopsis seedling root [24]. In addition to the *WUS* gene, the function of other *WOX* orthologues has been identified in many plants [25,26,27]. However, only a few *WOX* genes have been identified in cotton, the function of most of which remains unknown, especially in somatic embryogenesis. In the present study, we identified *WOX* genes in cotton using the updated reference genome and conducted a comprehensive analysis of the cotton *WOX* genes. The expression pattern and functional roles of cotton *WOXs* in somatic embryogenesis were analyzed. Our study provides useful information and sets up an important foundation for research into regeneration-related genes in cotton.

## 2. Results

### 2.1. Identification of WOX Genes in Cotton

A total of 40 *WOX* genes were identified in upload cotton. The result was almost consistent with previous studies [28]. In addition, 15, 24, 31, 13, 21, and 14 *WOX* gene members were identified in Arabidopsis, tobacco, soybean, rice, maize, and wheat, respectively. All these WOXs contain a complete HD. In cotton, at the C terminal region, 4 have EAR motif and 24 have the so-called WUS-box domain (Figure 1, Table 1).

### 2.2. Phylogeny Analysis of Cotton WOX Genes

Phylogenetic tree was constructed using the maximum likelihood (ML) method and showed that WOXs in upland cotton were divided into three clades: ancient, intermediate, and modern clades as convention. The modern clade contains the majority of cotton WOXs, which consists of 24 (60.0%) cotton WOXs (Figure 2A, Table 1). As for Arabidopsis, only about half of the *AtWOXs* are clustered in the modern clade, implying the expansion and diversification of the modern clade in cotton. Both ancient and intermediate clades contain eight WOXs (Figure 2A, Table 1).

### 2.3. Gene Structure and Conserved Amino Acid Motif Analysis of Cotton WOX Genes

The gene structure of all cotton *WOXs* was illustrated and showed that almost all cotton *WOX* genes have one or two introns, except for four cotton *WOX* genes in the modern clade which have three introns. Though the gene structure varies across different clades, genes in the same subclades showed the same gene structure (Figure 2B), implying that the phylogeny tree was reliable.

Six motifs were identified in cotton WOXs in our study (Appendix A). Motif 1 and motif 2 were located in the HD; almost all cotton WOXs contain motif 1 and motif 2 simultaneously, except for *GH_A11G3685*, which contains only motif 2. Motif 5 corresponds to the WUS-box motif and exists in all WOXs from the modern clade. Motif 3 exists only in the ancient clade. Motif 4 can only be found in the intermediate clade. Motif 6 is located in the ending of the C-terminal and exists in eight of the cotton WOXs from the modern clade (Figure 2C). The specific distribution of different motifs may confer the unique function of different genes.

### 2.4. Evolutionary Analysis of WOX Gene Family in Core Seed Plants

To investigate the evolution of *WOX* genes in angiosperm plants, all WOX proteins in core model angiosperm plants (including four dicots, Arabidopsis, cotton, tobacco, and soybean, and three monocots: rice, maize, and wheat) were used to construct the ML tree. As occurred in previous studies, all these WOXs were divided into ancient, intermediate, and modern clades (Figure 3).

In the ancient clade, the phylogeny branches of WOXs from monocots and dicots were mixed, suggesting that the ancient clade emerged at a more ancient timepoint before the divergence of monocots and dicots [14]. Meanwhile, in the intermediate and modern clades, WOXs from monocots and dicots were distinctly separated into two subclades, implying the divergent evolution of WOXs alongside the evolution of monocots and dicots.

### 2.5. Collinearity and Duplication Analysis of Cotton WOX Genes

Based on the genes coordinate annotation data, we mapped the identified upland cotton *WOX* genes on chromosomes. The cotton *WOX* genes were distributed unevenly across the majority of chromosomes, except for chrA4, chrA6, and chrA9 and chrD4, chrD6, and chrD9. Each chromosome contains one to three *WOX* genes (Figure 4).

Collinearity analysis showed that the cotton *WOX* genes belonged to segmental duplicates resulting from polyploidy and chromosome rearrangements. There were 46 homologous gene pairs involving all *WOX* genes located in collinearity blocks. Of the 46 homologous gene pairs, 20 homologous gene pairs involving 40 genes were orthologous homologous pairs between subgenome A and subgenome D. The other 13 homologous gene pairs were derived from segmental duplication, with 6 segmental duplicates gene pairs and 7 segmental duplicates gene pairs in the A and D subgenome, respectively. The majority (10) of which involved genes from the modern clade, except for 2 homologous gene pairs involving the intermediate clade and 1 homologous gene pair involving the ancient clade. Others were genes with homologs in other subgenome. No tandem duplication was observed in upland cotton *WOX* genes (Figure 4). This implied that segmental duplication extensively existed in the modern *WOX* clade in cotton.

### 2.6. qRT-PCR Validation of RNA Sequencing

In the present study, we focused on the expression pattern of *WOX* genes during cotton somatic embryogenesis. We previously conducted RNA sequencing across the entire process of upland cotton somatic embryogenesis, which initiated from hypocotyl dedifferentiated into non-embryogenic callus (NEC) and then transformed into embryogenic callus (EC) and differentiated and developed into global embryo (GE), torpedo-shaped embryo (TE), and cotyledon embryo (CE) sequentially to obtain the gene expression profiles during this developmental process. Nine differentially expressed genes, including five *WOX* genes, were selected to validate the expression profiles derived from RNA sequencing through qRT-PCR (specific qRT-PCR primers are listed in Appendix A). The results showed that all the tested genes showed comparable expression patterns between the expression levels from qRT-PCR and RNA sequencing analysis (Figure 5); the expression patterns derived from the two methods were highly correlated, with correlation coefficient (R^2^) values range from 0.87 to 0.94 (all *p*-values < 0.01) (Figure 5), suggesting the reliability of RNA sequencing.

### 2.7. Expression Pattern of Cotton WOX Genes in Somatic Embryogenesis

The expression pattern of *WOX* genes showed that about a half of the cotton *WOX* genes (19) were almost non-expressed in cotton somatic embryogenesis, including 13 *WOX* genes with TPM less than one and 6 *WOX* genes with TPM less than three, as shown in cluster IV. The other half (21) of the cotton *WOX* genes were actively expressed with relatively high expression levels of TPM > 5 during somatic embryogenesis, as shown in cluster I-III. Clustering based on expression level showed that these actively expressed *WOX* genes were divided into three clusters (cluster I, cluster II, cluster III) (Figure 6). The *WOX* genes in cluster I were actively expressed at different early stages (stages T1–T5) of somatic embryogenesis as the cells and tissues remained non-embryogenic, while they were expressed in a down-regulated manner obviously at late stages (stages T6–T9, except for stage T7) when the tissues were transformed to be embryogenic, even though some of them showed themselves to be anomalously expressed at a high level at the T7 stage, especially *GH_A05G2053*. This suggested that the high expression level of *WOX* genes in cluster I may be not good for the acquisition of embryogenic ability and late embryo development. The *WOX* genes in cluster II were obviously specifically expressed at a high level in embryogenic tissues (stages T6–T9). For *JH713*, the expression of these *WOX* genes in cluster II were significantly up-regulated when tissues were transformed into embryogenic ones, especially *GH_D05G1617* and *GH_A05G1589* (*GhWOX9*), implying that they play important roles in the acquisition of embryogenic ability and late somatic embryo development (stages T6–T9) in cotton somatic embryogenesis. The *WOX* genes in cluster III were expressed only at early callus induction stages (stages T1–T3), suggesting that these cluster III genes may participate in the early dedifferentiation processes of somatic embryogenesis.

For different phylogenetic clads, it was notable that the majority of cotton *WOX* genes from the modern clade were not expressed in somatic embryogenesis, with only about 40% (10) of *WOX* genes from the modern clade being expressed. Half of the *WOX* genes (four) from the ancient clade were actively expressed in different stages of cotton somatic embryogenesis, while almost all (seven) *WOX* genes from the intermediate clade (except for *GH_A10G0308*) showed active expression in cotton somatic embryogenesis. In all the actively expressed *WOX* genes, cluster I consisted of four *WOX* genes from the ancient clade and four *WOX* genes from the modern clade; cluster II consisted of three *WOX* genes from the intermediate clade and four *WOX* genes from the modern clade; and cluster III consisted of four *WOX* genes from the intermediate clade and two WOX genes from the modern clade. All the actively expressed *WOX* genes from the ancient clade belonged to cluster I. The actively expressed *WOX* genes from the intermediate clade were distributed in both cluster II and cluster III, as cluster II and cluster III represent different expression patterns and were actively expressed in late stages and early stages of somatic embryogenesis, respectively, suggesting the functional diversification of *WOX* genes from the intermediate clade. So, the modern clade *WOX* genes were distributed in cluster I, cluster II, and cluster III.

### 2.8. Cis-Acting Regulatory Elements Analysis of Cotton WOX Genes’ Promoters

By searching the promoter region sequences (1.5 kb sequence upstream of start codon) against the plantCARE database, the diversity of cis-acting regulatory elements was detected. As a result, phytohormone response, stresses response, and some other related cis-acting regulatory elements were widespread in the promoter regions of cotton *WOX* genes (Figure 7; Appendix A). As for stress response, all cotton *WOX* genes had light response cis-acting regulatory elements, and more than 85% of cotton *WOX* genes had heat, osmotic stress, low pH, and nutrient starvation stress response regulatory elements in their promoter regions. Anoxic stresses response cis-acting regulatory elements were found in the promoter regions of 80% of cotton *WOX* genes. Wounding/pathogen and drought response cis-acting elements were found in the promoter regions of about 70% (72.5% and 70.0%, respectively) of cotton *WOX* genes. Cold response and water stress response cis-acting elements existed in 27.5% and 17.5% of cotton *WOX* genes’ promoter regions, respectively. As for phytohormone response, all cotton *WOX* genes had various hormone response cis-acting elements in their promoter regions. The ABA response cis-acting element was found most generally widespread in cotton *WOX* genes’ promoter regions, followed by the ETH response and GA response cis-acting element, which existed in 70%, 65%, and 60% of cotton *WOX* genes’ promoter regions, respectively. GA response and SA response cis-acting elements existed in 42.5% of cotton *WOX* genes’ promoter regions. A total of 22.5% of cotton *WOX* genes had auxin response cis-acting elements in their promoter regions. Additionally, Cd response, cell proliferation, flavonoid biosynthetic regulation, endosperm expression, meristem expression, zein metabolism regulation, and some others cis-acting elements were found in some of the cotton *WOX* genes’ promoters.

### 2.9. Gene Co-Expression Analysis of Cotton WOX Genes during Somatic Embryogenesis

To further explore the roles and expression network of *WOX* genes in cotton somatic embryogenesis, we conducted weighted gene co-expression network analysis (WGCNA) based on the data of previously conducted RNA sequencing of upland cotton somatic embryogenesis. WGCNA classified all the differentially expressed genes into several co-expression gene modules. The result showed that most cotton *WOX* genes that were expressed in somatic embryogenesis were located in the module, which showed a significant positive correlation with embryogenic tissues (cor. = 0.59, *p*-value < 0.01.). For this embryogenic-tissue-associated module, cotton *WOX* genes involved in the co-expression network were constructed and displayed (Figure 8A). The result showed that seven *WOX* genes (*GH_D13G2554*, *GH_A13G2563, GH_D05G1617*, *GH_A07G1065*, *GH_D10G0320*, *GH_D07G1052*, *GH_A05G1589*) corresponding to cluster II of the above expression clustering heatmap (Figure 5) were co-expressed with the other 1881 genes with a weight value co-expression > 0.5 as a criterion. GO enrichment showed that for these WOX co-expressed genes, with the molecular function of DNA helicase activity, DNA replication origin binding, microtubule motor activity, microtubule binding, lipid binding, and the biological process of DNA replication, DNA repair, lipid storage, and seed oil body biogenesis were enriched (Figure 8B). The KEGG pathway annotation showed that these co-expressed genes were involved in several environmental information processing pathways, mainly as plant hormone signal transduction, MAPK signaling pathway, plant–pathogen interaction, the phosphatidylinositol signaling system, and ABC transporters; a variety of genetic information processing pathways, especially as DNA replication, RNA transport, homologous recombination, protein processing and proteolysis; and some primary and secondary metabolism pathways, especially amino acids metabolism, carbon metabolism, and fatty acid metabolism (Appendix A). The KEGG pathway enrichment analysis found that DNA replication, DNA repair, homologous recombination, fatty acid metabolism, plant hormone signal transduction pathways, valine, leucine, and isoleucine biosynthesis, pantothenate and CoA biosynthesis, and cutin, suberine, and wax biosynthesis were the most significantly enriched (Figure 8C).

According to the co-expressed network, clustering showed that about 77 genes were located at the hub position. All these hub genes were co-expressed with at least four *WOX* genes simultaneously. In detail, 10 genes were co-expressed with seven *WOX* genes simultaneously, 24 genes were co-expressed with six *WOX* genes simultaneously, and 18 and 15 genes were co-expressed with five and four *WOX* genes simultaneously, respectively. GO enrichment analysis showed that the molecular function of lipid binding and DNA binding and the biology process of the regulation of transcription, DNA replication, amino acid metabolism, lipid transport, and proton transport were the most highly enriched for these hub co-expressed genes (Figure 8A). KEGG pathway annotation showed that these hub co-expressed genes were involved in similar pathways, as described above for all the co-expressed genes, while enrichment analysis of these hub co-expressed genes showed that valine, leucine, and isoleucine biosynthesis, cutin, suberine, and wax biosynthesis, fatty acid metabolism, pantothenate and CoA biosynthesis, anthocyanin biosynthesis, and ABC transporters pathways were significantly enriched, which was obviously different from the whole co-expressed genes set (Figure 8A).

Except for the above embryogenic-tissues-associated module, some *WOX* genes were classified into other two modules in WGCNA, as corresponding to cluster I and cluster III of the above expression clustering heatmap, respectively. The co-expression network was constructed and displayed, respectively (Appendix A). For the co-expressed genes with cluster I *WOX* genes, GO enrichment analysis showed that the chloroplast- and photosystem-related cellular component, the molecular function of chlorophyll, tetrapyrrole binding, electron transfer activity, and photosynthesis-related biology process were enriched. KEGG pathway annotation showed that the photosynthesis, carbon metabolism, glutathione metabolism, and plant hormone signal transduction were the most involved pathways (Appendix A). For the co-expressed genes with cluster III *WOX* genes, GO enrichment analysis showed that for the molecular function of polygalacturonase activity and the biology process of the carbohydrate metabolic process, cell wall organization was enriched. KEGG pathway annotation showed that the most co-expressed genes were involved in pentose and glucuronate interconversions and RNA transport pathways (Appendix A).

## 3. Discussion

The plant-specific transcription factor WOXs family plays an important role in a variety of plant developmental processes, including embryo development as well as somatic embryogenesis [29]. It has been proved that some WOX members are vital regulators in plant somatic embryogenesis, such as *AtWUS* [30], *ZmWUS2* [17], *TaWOX5* [16], and so on, and significantly improve the efficiency and shorten the period of somatic embryogenesis, even in many other related species. In cotton, *GhWUS*, a homolog of *AtWUS*, has been reported to improve somatic embryogenesis, while other members remain uninvestigated [24]. In this study, we conducted a comprehensive investigation and characterization of the cotton *WOX* gene family in somatic embryogenesis through genome-wide analysis. As an allotetraploid species, cotton contains over twice the amount of *WOX* gene members than its close relative Arabidopsis, suggesting the expansion of the *WOX* gene family in cotton. Analysis showed that no tandem duplication was identified; segment duplication may greatly contribute to the expansion of the *WOX* gene family. Phylogenetic analysis showed that WOX members of mono and dicot species were clustered into a different group distinctly, except for ancient clade in which mono and dicot species were mixed in a subclade. This was consistent with a previous study which showed that the ancient clade of WOXs was more ancient than other members; the intermediate clade and modern clade of WOX members emerged after the divergency of monocots and dicots [7,30]. It was shown that cotton WOXs have a close relation with Arabidopsis ones. Each Arabidopsis WOX containing a subclade contained at least twice the amount of WOX family members from cotton, especially in the modern clade, in which one Arabidopsis WOX member may correspond to even four cotton WOX members in one subclade. The expansion of the modern clade of the cotton *WOX* gene family may provide potential for a more diverse gene function through gene function divergency or sustain the functional stability of *WOX* modern clade genes.

Expression pattern analysis showed that only about half of the cotton *WOX* gene members were actively expressed in somatic embryogenesis, while the other half of the gene members showed no expression, even including the homologs of *AtWUS*. Overall, orthologous *WOX* genes from different subgenome showed a similar expression pattern (Figure 5), which was consistent with the similar cis-acting elements harbored in their promoters (Figure 6). It is worth noting that the actively expressed *WOX* genes in somatic embryogenesis showed three different expression patterns, implying that different *WOX* genes play roles in different stages of somatic embryogenesis. The cotton *WOX* genes in cluster I were actively expressed in almost the whole process of somatic embryogenesis, especially for homologs of *AtWOX13* (*GH_D03G0016*, *GH_A02G2045*, *GH_A07G1886*, *GH_D07G1885*). It has been reported that *AtWOX13* was required for callus growth and participated in floral transition from vegetative growth to reproductive growth and root development [31]. We speculated that cotton *WOX13* homologous genes may paly basic and versatile roles in somatic embryogenesis by participating in callus formation and growth at early stages and embryogenic differentiation and embryo development at late stages. As AtWOX13 is the only WOX member lacking a canonical WUS-box domain which is necessary for stem cell regulation and maintenance function [9], as does the homologs in cotton, we assumed that these *WOX13* homologous members may take part in some basic biological processes involved in somatic embryogenesis, though they may not have the stem cell regulation function. This was consistent with the result that the corresponding co-expressed gene set harbors the pathway annotation of photosynthesis, carbon metabolism, and glutathione metabolism [32]. In addition, four homologs of *AtWOX4* [15,33] (*GH_A05G2053*, *genegh_D05G2084*, *GH_D01G1267*, *GH_A01G1199*) showed a somewhat different expression pattern; these gene members were actively expressed in all non-embryogenic stages (T0–T5), while they were down-regulated significantly in the subsequent embryonic stages, except for the global embryo stage (T7), where the gene *GH_A05G2053* was the most representative. The markedly down-regulated expression of these *WOX4* homologous genes in embryogenic tissues implies that they may be unnecessary for embryogenic differentiation at the late stage of somatic embryogenesis. It was reported that *WOX4* participated in the procambial development in many plants [26,34]; the paradoxica up-regulated expression of these *WOX4* homologous genes at the global embryo stage may attribute to the procambial development of global embryo. Cotton *WOX* genes in cluster II were specifically expressed in embryogenic tissues and embryo (stages T6–T9) during cotton somatic embryogenesis. Phylogeny analysis showed that these *WOX* members were homologs of *AtWOX8/9* (*GH_A05G1589*, *GH_D05G1617*, *GH_D10G0320*) and *AtWOX2* (*GH_A07G1065*, *GH_D07G1052*, *GH_D13G2554*, *GH_A13G2563*). As both *WOX8/9* and *WOX2* had been reported to function in embryo patterning [14,35], the high expression level of cluster II may also reflect that these *WOX* genes deeply participated in and promoted embryo patterning in cotton somatic embryogenesis. Embryo patterning and development involves extensive genetic reprogramming and diverse biological processes; the co-expression analysis of these *WOX8/9* and *WOX2* homologous genes in cluster II showed that DNA replication, DNA repair, homologous recombination, fatty acid metabolism, and valine, leucine, and isoleucine biosynthesis, pantothenate and CoA biosynthesis, and cutin, suberine, and wax biosynthesis were the most significantly enriched in embryo patterning during cotton somatic embryogenesis. The MAPK signaling pathway, phosphatidylinositol signaling system, and ABC transporters were also involved. Cotton *WOX* genes in cluster III, which consisted of the homologous genes of *WOX11*/*12* and *WOX5*/*7*, were only actively expressed at the early stage of callus formation and growth, implying their roles in callus initiation and growth. Overall, of all these cotton *WOX* genes, the homologous genes of *WOX2*, *WOX4*, *WOX5/7*, *WOX8/9*, *WOX11/12*, and *WOX13* were actively expressed at different stages during cotton somatic embryogenesis, suggesting that they may function in different developmental stages. *GhWOX5/7* and *GhWOX11/12* may function in callus initiation and growth. *GhWOX13* may participate in some basic biological and metabolism processes and function at all stages throughout somatic embryogenesis. *GhWOX4* may specifically participate in procambial development at the early stage of somatic embryo development. *GhWOX2* and *GhWOX8/9* may function in somatic embryo patterning and embryo development during cotton somatic embryogenesis. As the difference in regeneration ability between the different genotypes of cotton cultivars mainly derived from the embryogenic transition and somatic embryo formation, rather than callus initiation and growth, according to the expression pattern clustering, we think that the most representative *WOX4* homologous gene GH_A05G2053 and *WOX13* homologous genes *GH_D03G0016* and *GH_A02G2045* in cluster I as well as *WOX8/9* homologous genes *GH_A05G1589* and *GH_D05G1617* in cluster II are worthy of further functional research.

Interestingly, except for the *WOX4* homologous gene GH_A05G2053, the other four most representative genes, as *WOX13* homologous genes of *GH_D03G0016* and *GH_A02G2045*, and *WOX8/9* homologous genes of *GH_A05G1589* and *GH_D05G1617*, described above were all auxin response, as the auxin response cis-acting element was harbored in their promoter regions. According to the cis-acting elements analysis, there were another five *WOX* genes which had auxin response cis-acting regulatory elements, while they were non-expressed during cotton somatic embryogenesis. As it has been widely proved that plant hormones, especially auxin, play an important role in cotton somatic embryogenesis, we speculate that *WOX* genes may be involved in cotton somatic embryogenesis partly through the auxin response activation of *GhWOX8/9* and *GhWOX13*, which may be an important part of the auxin signaling pathway during cotton somatic embryogenesis.

The acquisition of embryonic ability for callus is the most intractable difficulty faced in cotton somatic embryogenesis. The key stage of transformation from non-embryonic callus into embryonic callus was mostly focused on in the present study. We showed that *GhWOX8/9* and *GhWOX2* were markedly up-regulated expressed during the process transformation from non-embryonic callus to embryonic callus, implying that these genes may play vital roles in the key embryonic transition process. In Arabidopsis, maize, and some other plants, the *WUS* gene plays the function of promoting somatic embryogenesis [17,18,24], while as *GhWUS* was not expressed during cotton somatic embryogenesis, we speculate that these *GhWOX8/9* and *GhWOX2* genes may take up a similar role instead of *WUS*, or merely function in a redundant way.

## 4. Materials and Methods

### 4.1. Identification of WOX Genes in Upland Cotton

In the present study, the latest assembled genome and gene annotation of *G. hirsutum* was downloaded from a public website (http://cotton.zju.edu.cn/index.htm, accessed on 19 January 2023) as a reference genome [36]. To identify the *WOX* gene family members, similarity search (E-value < 1 × 10^−5^, identity > 50%) against the proteome of *G. hirsutum* using BLAST+ v.2.13.0 [37] with the amino acid sequences of Arabidopsis WOXs retrieved from TAIR database (https://www.arabidopsis.org/, accessed on 19 January 2023) was conducted [38]. The resulting protein hits from the BLAST search were used for subsequent filtering. The Pfam (http://pfam.janelia.org, accessed on 19 January 2023) [39] and SMART (http://smart.embl-heidelberg.de/, accessed on 19 January 2023) search tools [40] were used to identify and check the presence of characteristic domains. Items containing the “DNA-binding homeodomain (corresponding ID of PF00046 in Pfam or SM000389 in SMART)” domain were recognized as candidate WOXs. Multiple sequences alignment conducted by MUSCLE v.5.1.0 [41] was used to identify other functional domains, including the EAR domain and WUS-box.

### 4.2. Phylogenetic Analysis, Gene Structure, and Motif Analysis of Cotton WOX Genes

For phylogenetic analysis, amino acid sequences of *WOX* family genes identified in *G. hirsutum* and other representative seed plant species including Arabidopsis, tobacco, soybean, rice, maize, and wheat were used to perform multiple sequences alignment using MUSCLE v.5.1.0 [41]. Phylogenetic tree was constructed by IQTREE v.1.6.12 [42] using a maximum likelihood (ML) statistical method and best substitution model selected automatically. Coordinates of exon-intron of cotton *WOX* genes were extracted from gene annotation information. This information was submitted to GSDS v.2 (http://gsds.cbi.pku.edu.cn/, accessed on 15 March 2023) [43] to illustrate the map of gene structure. MEME (https://meme-suite.org/meme/, accessed on 21 March 2023) tools [44] were used to identify conserved motifs in protein sequences of cotton WOXs.

### 4.3. Genomic Distribution, Collinearity, and Duplication Analysis of Cotton WOX Genes

Genomic gene annotation information was used to map genes on chromosomes. MCScan [45] was used to identify collinearity blocks in *G. hirsutum*. Collinearity relations of *WOX* genes were displayed by circular figure plotted by Circos v.0.69 [46]. Genes in the same subgenome were recognized as derived from segmental duplication when located in collinear blocks.

### 4.4. Expression Pattern Analysis of Cotton WOX Genes during Somatic Embryogenesis

We previously conducted an RNA sequencing across the entire process of upland cotton cultivar *JH713* somatic embryogenesis which initiated from hypocotyl and developed into somatic embryo at the end. During this long process, we sampled a total of 10 timepoints: T0 represented the hypocotyl, T1–T4 represented dedifferentiated non-embryogenic callus (NEC), T5 represented the transitional stage from NEC into embryogenic callus (EC), T6 represented EC, and T7–T9 represented global embryo (GE), torpedo-shape embryo (TE), and cotyledon embryo (CE), respectively. RNA was isolated and used to construct a sequencing library and was then sequenced using the Illumina platform. This enabled us to investigate the expression profiles of *WOX* genes during cotton somatic embryogenesis. HISAT2 v.2.2.1 [47] was used to map the sequencing reads to the assembled *G. hirsutum* reference genome; StringTie v.2.2.0 [48] was used to perform genes quantification. Fragments per kilobase of exon per million reads mapped (TPM) values were computed to represent the gene expression levels. Heatmaps with hierarchy clustering of *WOX* genes were plotted using R software v.4.3.1 package pheatmaps based on log 10-transformed TPM values.

### 4.5. qRT-PCR Validation of the RNA Sequencing

To validate the expression profiles of differentially expressed genes derived from RNA sequencing, six genes, including *WOX* and *ZHD* genes, were selected for qRT-PCR validation. Specific qPCR primers were designed using the National Center for Biotechnology Information (NCBI) Primer-BLAST (https://www.ncbi.nlm.nih.gov/tools/primer-blast/index.cgi, accessed on 8 April 2023) and synthesized commercially (Sangon Biotech (Shanghai) Co., Ltd., Shanghai, China). Gene *GhHis3* (Genbank accession number: AF024716) was selected as an internal reference. cDNA was synthesized using the HiScriptIII1stStrandcDNASynthesis Kit (Vazyme Biotech Co., Ltd., Nanjing, China) according to the manufacturer’s instructions. Then, the cDNA templates were diluted five times prior to amplification. The 20 µL qPCR reaction system contained 1 µL cDNA templates, 10 µL 2× TransStart Top Green qPCR Super Mix (TransGen Biotech Co., Ltd., Beijing, China), 0.5 µL of each 10 µM forward and reverse primers, and 8 µL ddH2O. All qPCR reactions were performed in triplicate on a Roche Light Cycler 480 qPCR instrument. The thermal cycling program was pre-incubated at 95 °C for 5 min, followed by 40 cycles of denaturation at 95 °C for 10 s, annealing at 58 °C for 10 s, and extension at 72 °C for 10 s. Relative quantitation analysis was computed using the online tool “A shiny for the analysis of real-time PCR data” (https://ihope.shinyapps.io/qRT-PCR-Pipeline/, accessed on 16 April 2023).

### 4.6. Co-Expression Network Analysis

Based on the expression profiles of all differentially expressed genes during cotton somatic embryogenesis, we performed weighted gene co-expression network analysis (WGCNA) using R package WGCNA v.1.70-3 [49]. The constructed co-expression gene modules were further subjected to association analysis with different samples. The gene modules involving *WOX* genes were screened to displayed the co-expression network by VOSviewer v.1.6.17 [50]. GO term and KEGG pathway enrichment analysis using the hypergeometric test method was conducted using the R package clusterProfiler v.3.6.0 [51].

## 5. Conclusions

In cotton, *WOX* genes were identified genome-wide in the present study. All the 40 cotton *WOX* genes were classified into three clades, as occurred in previous studies. Dicot and monocot plant *WOX* genes were clearly clustered, except for in the ancient clade. The gene structures and motifs of cotton *WOX* genes were analyzed. Though the *WOX* gene family in cotton expanded in some degrees compared to Arabidopsis, only half of them were actively expressed during somatic embryogenesis. Different *WOX* gene members may function in different stages during cotton somatic embryogenesis according to the expression pattern. The most representative *GhWOX8/9*, *GhWOX4*, and *GhWOX13* were potential candidate genes that may play important roles in cotton somatic embryogenesis. Co-expression analysis illustrated the involved biological processes during somatic embryogenesis. The comprehensive analysis of the cotton *WOX* gene family and its roles in cotton somatic embryogenesis provided useful information for relative functional research. The specific roles of candidate *WOX* genes mentioned above may be worthy of explication by further experimental research.

## Figures and Tables

**Figure 1 ijms-24-11077-f001:**
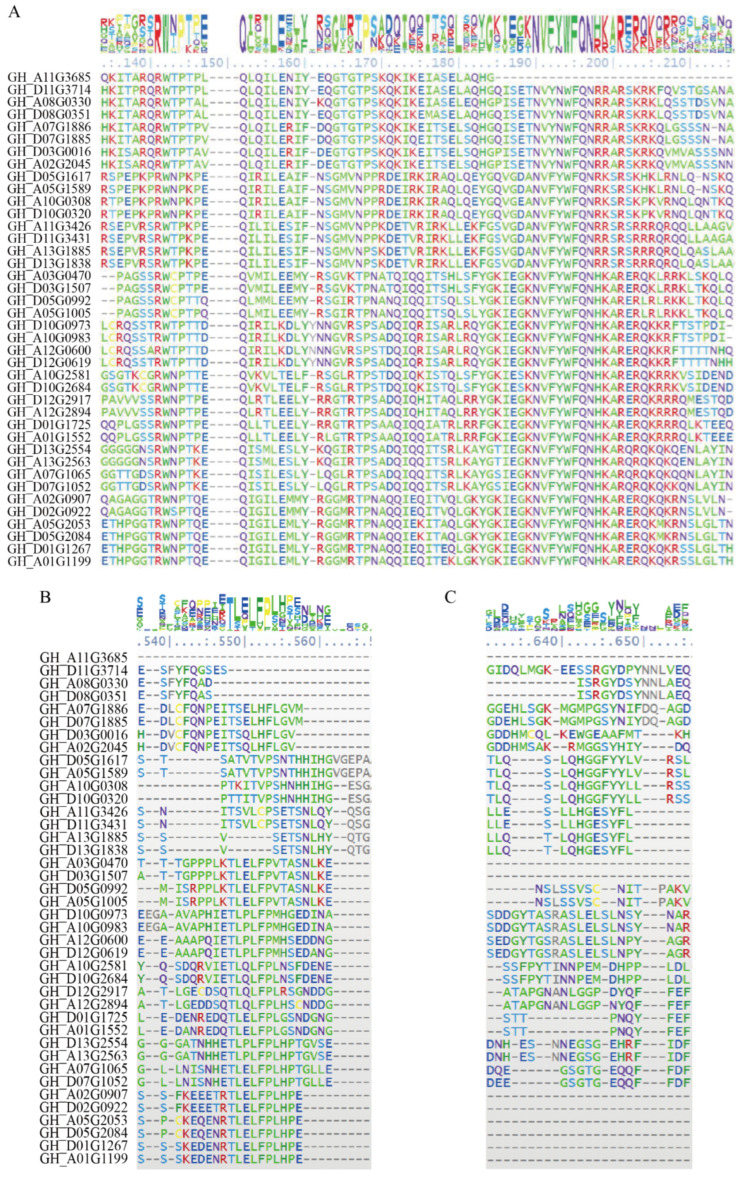
Multiple sequences alignment of the protein sequences of cotton WOXs showed three major functional domains: HD (**A**), WUS-box (**B**), and EAR motif (**C**).

**Figure 2 ijms-24-11077-f002:**
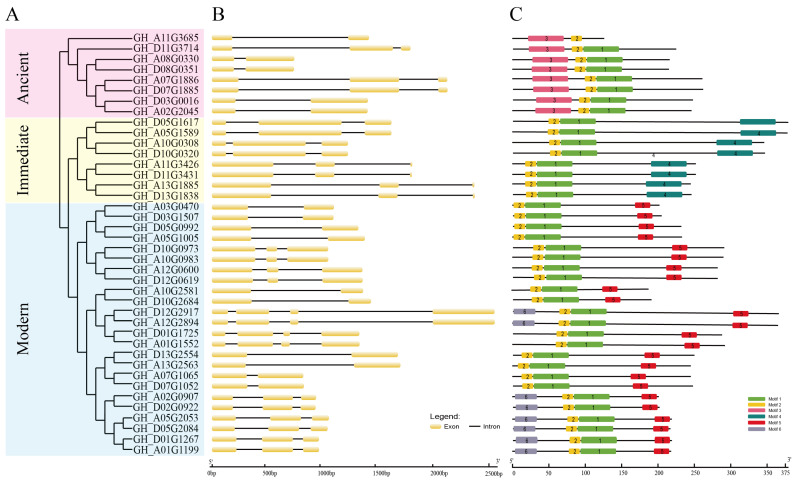
Phylogenetic tree of cotton WOXs constructed using the maximum likelihood (ML) method based on full-length amino acid sequences. (**A**) shows that cotton WOXs were classified into ancient, intermediate, and modern clades. Gene exon-intron organization structure of cotton WOX family genes (**B**) and identified motifs in cotton WOXs (**C**).

**Figure 3 ijms-24-11077-f003:**
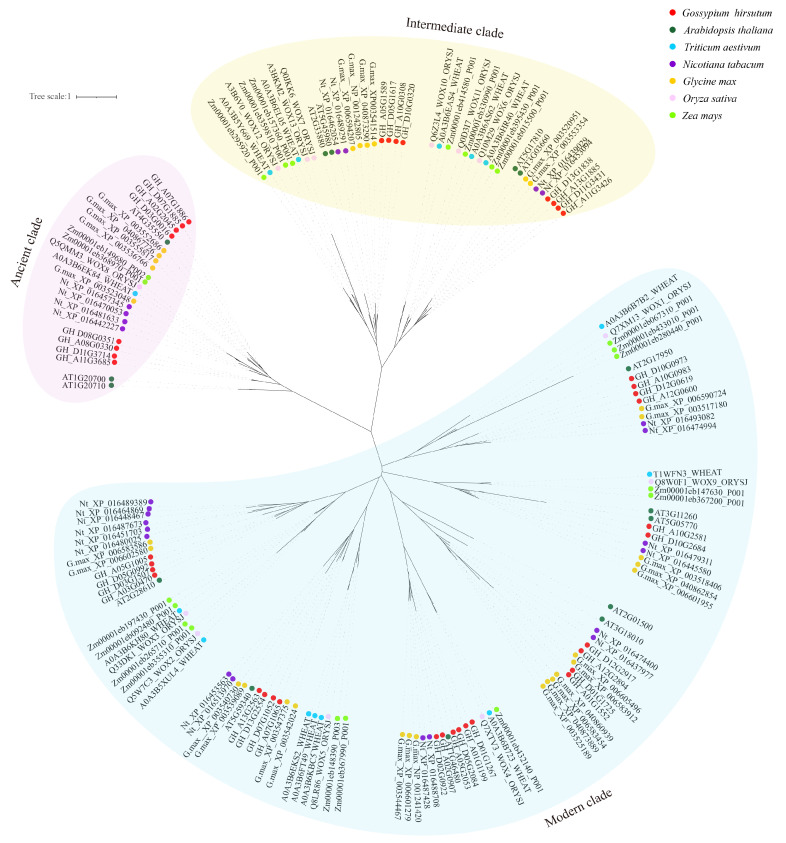
Phylogenetic tree of identified WOXs from cotton and representative core model plant species from angiosperm, including monocots (rice, maize, wheat) and dicots (Arabidopsis, cotton, tobacco, soybean). The tree was constructed using the maximum likelihood (ML) method based on the full-length amino acid sequences of WOXs.

**Figure 4 ijms-24-11077-f004:**
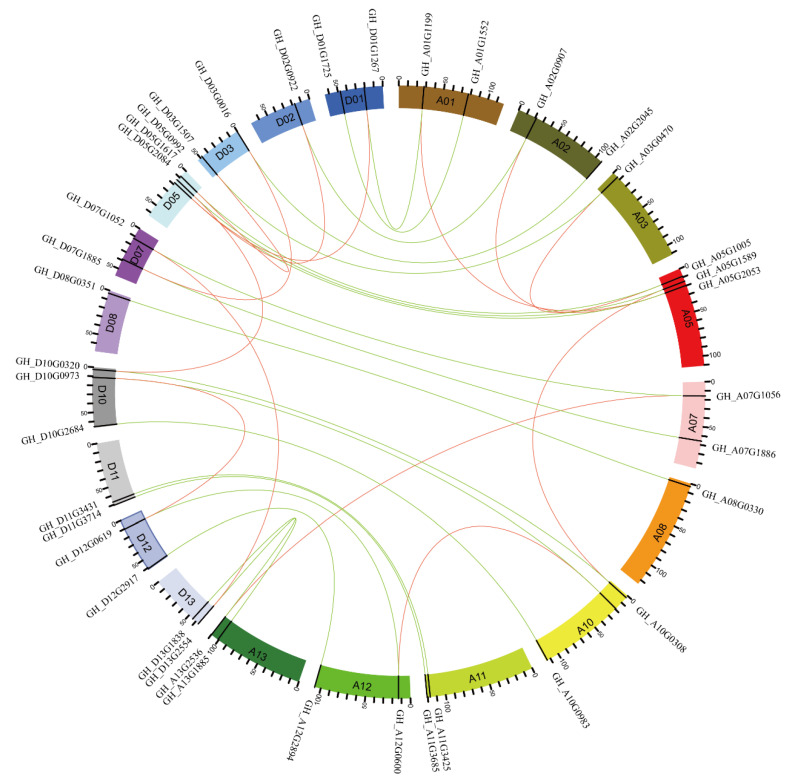
Chromosomal distribution and collinearity of cotton *WOX* genes indicated gene duplications. Genes were displayed by stripes on chromosomes. Chromosomal segmental duplication regions were analyzed using MCScan. Homologous *WOX* gene pairs in segmental duplication regions are linked with red-colored curves; orthologous *WOX* gene pairs between subgenome A and subgenome D are linked with green-colored curves.

**Figure 5 ijms-24-11077-f005:**
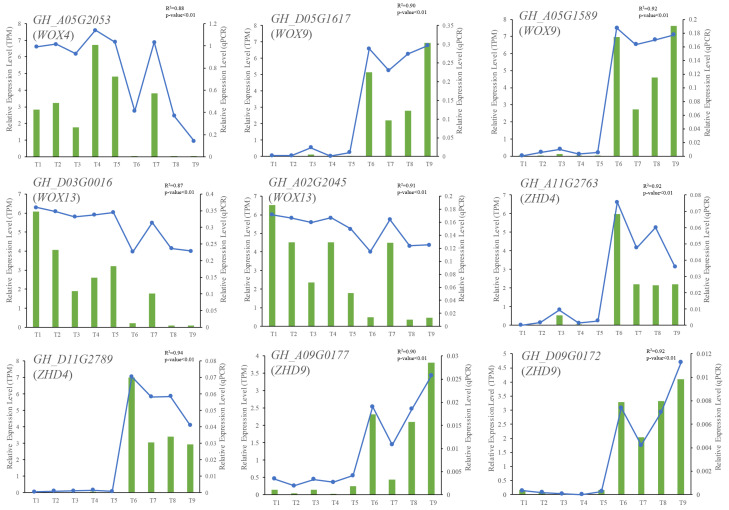
Validation of the expression profiles by qRT-PCR. Nine differentially expressed genes, including five *WOX* genes (*GH_A05G2053*, *GH_D05G1617*, *GH_A05G1589*, *GH_A02G2045,* and *GH_D03G0016*), were selected to conduct qRT-PCR. The horizontal axis showed different timepoints of cotton somatic embryogenesis by T1 to T9, representing NEC (T1-T4), NEC/EC (transitional stage from NEC to EC) (T5), EC (T6), GE (T7), TE (T8), and CE (T9), respectively. The green columns represent the gene relative expression levels calculated by qRT-PCR; the dot-lines represent those calculated by TPM values.

**Figure 6 ijms-24-11077-f006:**
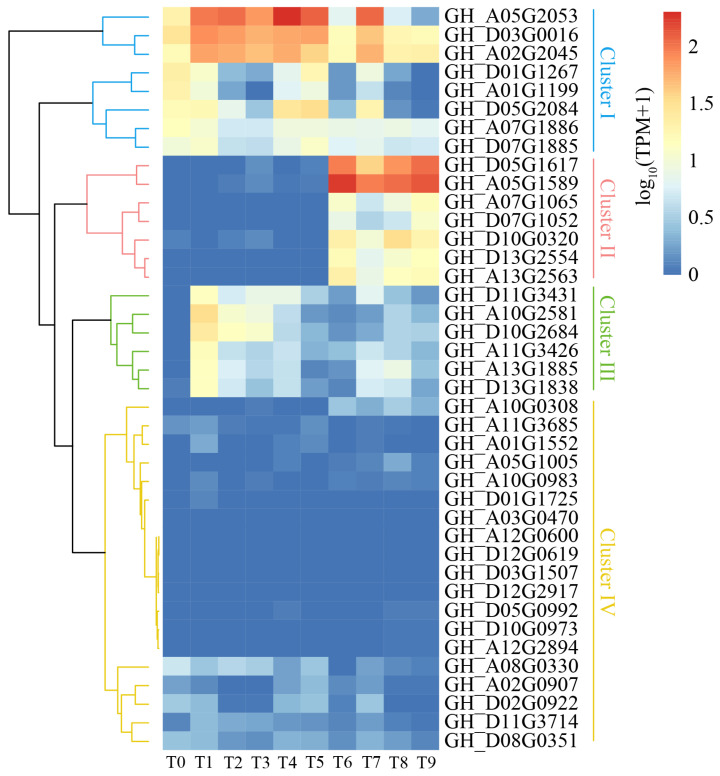
Expression profiles of cotton *WOX* genes during somatic embryogenesis which initiated from the timepoint of T0 to T9, representing hypocotyl (T0), NEC (T1–T4), NEC/EC (transitional stage from NEC to EC) (T5), EC (T6), GE (T7), TE (T8), and CE (T9), respectively. The heatmap was drawn based on log_10_-transformed TPM values of genes derived from the RNA sequencing analysis.

**Figure 7 ijms-24-11077-f007:**
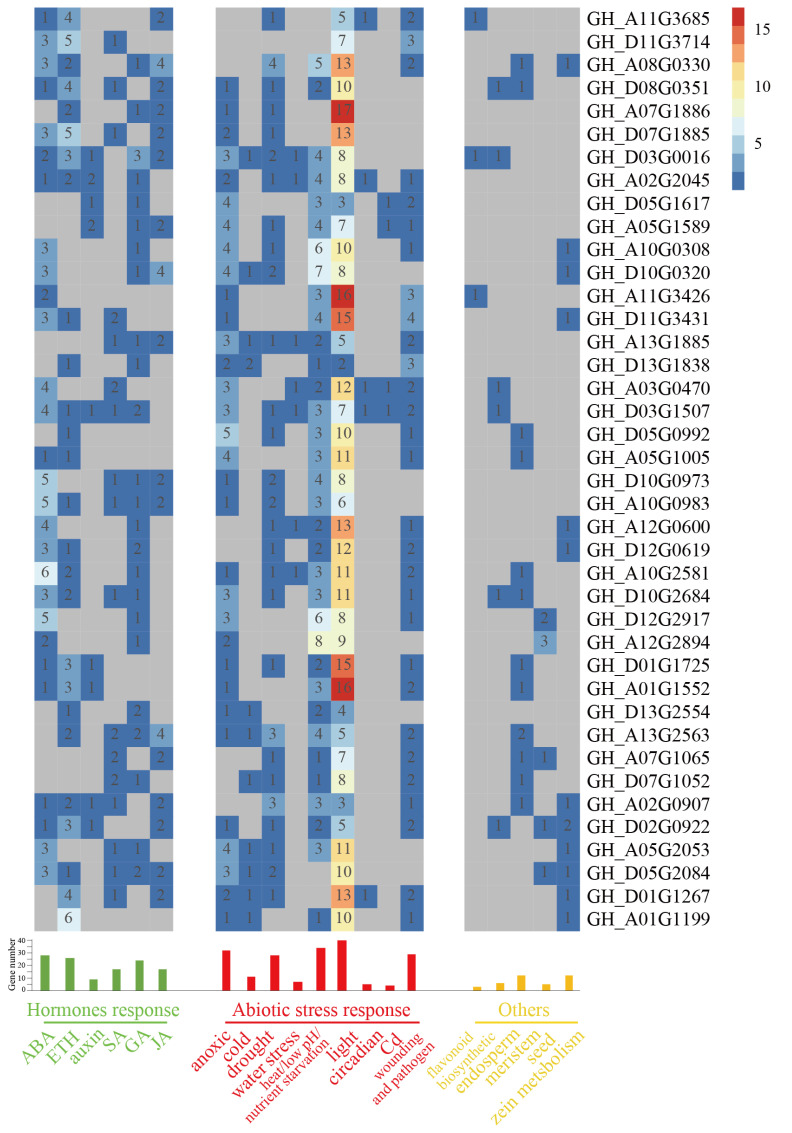
Statistics and classification of Cis-acting regulatory elements in cotton *WOX* genes’ promoters. Numbers in colored grids represented the numbers of specific cis-acting regulatory elements. Numbers on the tips of bars showed the total numbers of cotton *WOX* gene containing specific cis-acting elements.

**Figure 8 ijms-24-11077-f008:**
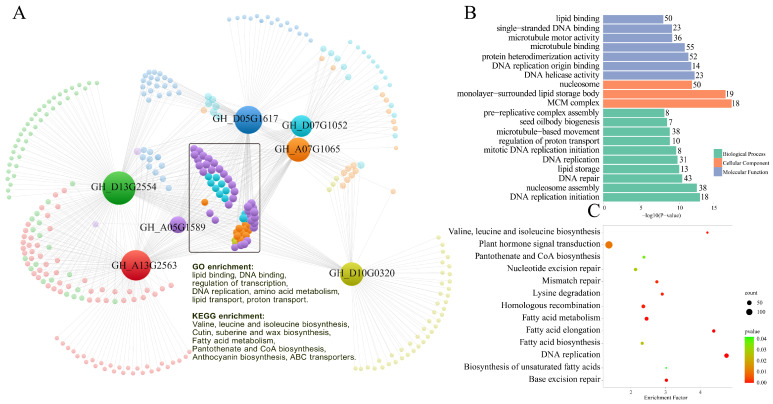
Co-expression network of cotton *WOX* genes during somatic embryogenesis based on the embryogenic-tissue-associated module in WGCNA results (**A**). Genes co-expressed with cotton *WOX* genes are represented by circle nodes in different colors; lines linking two nodes represent co-expression relationships between them. The size of the nodes represents the numbers of *WOX* genes that showed a co-expression relationship with the specific gene. The hub co-expressed genes are marked as a black tangle. The GO term (**B**) and KEGG pathway (**C**) enrichment analysis results of genes co-expressed with cotton *WOX* genes.

**Table 1 ijms-24-11077-t001:** Summary of cotton WOX protein domains.

Gene ID	Gene Name	Domains	Clade	Gene ID	Gene Name	Domains	Clade
*GH_A01G1199*	*WOX4*	WUS-box/HD	Modern	*GH_D01G1267*	*WOX4*	WUS-box/HD	Modern
*GH_A01G1552*	*WOX1*	WUS-box/HD	Modern	*GH_D01G1725*	*WOX1*	WUS-box/HD	Modern
*GH_A02G0907*	*WOX4*	WUS-box/HD	Modern	*GH_D02G0922*	*WOX4*	WUS-box/HD	Modern
*GH_A02G2045*	*WOX13*	HD	Ancient	*GH_D03G0016*	*WOX13*	HD	Ancient
*GH_A03G0470*	*WOX3*	WUS-box/HD	Modern	*GH_D03G1507*	*WOX3*	WUS-box/HD	Modern
*GH_A05G1005*	*WOX3*	WUS-box/HD	Modern	*GH_D05G0992*	*WOX3*	WUS-box/HD	Modern
*GH_A05G1589*	*WOX9*	HD	Intermediate	*GH_D05G1617*	*WOX9*	HD	Intermediate
*GH_A05G2053*	*WOX4*	WUS-box/HD	Modern	*GH_D05G2084*	*WOX4*	WUS-box/HD	Modern
*GH_A07G1065*	*WOX2*	WUS-box/HD	Modern	*GH_D07G1052*	*WOX2*	WUS-box/HD	Modern
*GH_A07G1886*	*WOX13*	HD	Ancient	*GH_D07G1885*	*WOX13*	HD	Ancient
*GH_A08G0330*	*WOX8*	HD	Ancient	*GH_D08G0351*	*WOX8*	HD	Ancient
*GH_A10G0308*	*WOX9*	HD	Intermediate	*GH_D10G0320*	*WOX9*	HD	Intermediate
*GH_A10G0983*	*WUS*	WUS-box/HD/EAR	Modern	*GH_D10G0973*	*WUS*	WUS-box/HD	Modern
*GH_A10G2581*	*WOX5*	WUS-box/HD/EAR	Modern	*GH_D10G2684*	*WOX5*	WUS-box/HD	Modern
*GH_A11G3426*	*WOX11*	HD	Intermediate	*GH_D11G3431*	*WOX11*	HD	Intermediate
*GH_A11G3685* *GH_A12G0600*	*WOX8* *WUS*	HD WUS-box/HD/EAR	Ancient Modern	*GH_D11G3714* *GH_D12G0619*	*WOX8* *WUS*	HD WUS-box/HD/EAR	Ancient Modern
*GH_A12G2894*	*WOX1*	WUS-box/HD	Modern	*GH_D12G2917*	*WOX1*	WUS-box/HD	Modern
*GH_A13G1885*	*WOX11*	HD	Intermediate	*GH_D13G1838*	*WOX11*	HD	Intermediate
*GH_A13G2563*	*WOX2*	WUS-box/HD	Modern	*GH_D13G2554*	*WOX2*	WUS-box/HD	Modern

## Data Availability

Not applicable.

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
