# Peer review of "Identification and Evolutionary Analysis of Cotton (Gossypium hirsutum) WOX Family Genes and Their Potential Function in Somatic Embryogenesis"

_ijms, 2023, doi:10.3390/ijms241311077_

Round 1
Reviewer 1 Report
-
What are the rationale/ reasons to use GhHis3 for internal reference of the qRT-PCR? Are there any other candidates for reference genes? Are the mediums of tissue culture supplemented with auxin or cytokinin when the callus tissue was sampled for expression profile?
-
Higher resolution diagram is necessary in particular for Figure 8. Some text in the figure was not clear.
-
What are the exact cis-regulatory elements that were identified in the promoter region of specific cotton WOX gene in response to auxin or cytokinin treatment? What is the proposed molecular mechanism?
The manuscript “Identification and evolutionary analysis of cotton (Gossypium hirsutum) WOX gene family and functional roles analysis in somatic embryogenesis” described the analysis of WOX genes in cotton through gene structure, phylogenetic tree and collinearity analysis. Further, qRT-PCR, RNA-sequencing and co-expression studies were performed to have a comprehensive understanding of the role of WOX gene family during the process of somatic embryogenesis in cotton.
Minor errors
Page 1, Line 11, Correspondence:, repeated twice, please drop one
Page 1, Line 17, intermediate instead of immediately, and modern clades, please be consistent throughout the manuscript
Page 1, Line 33, homologous, lower case letter
Page 1, Line 36, valine, cutin, lower case letter
Page 2, Line 48, were difficult to regenerate instead of regeneration
Page 2, Line 48-49, and the current established regeneration system in some cultivars
Page 2, Line 56, identified to play critical roles
Page 2, Line 68, showed that the WUS box, the should be lower case letter
Page 2, Line 73, please italicize the scientific name Arabidopsis thaliana
Page 2, Line 86-87, but this did not work in reluctant cultivars
Page 2, Line 92, using the updated reference genome, grammar error
Page 2, Line 94, provides useful information and sets up important foundation for regeneration related genes in cotton researches
Page 3, Line 99-100, maize and wheat respectively
Page 5, Line 108, was constructed with Maximum Likelihood (ML) method and showed that woxs…..
Page 5, Line 109, intermediate, please be consistent throughout the manuscript
Page 5, Line 112, only about half AtWOXs are clustered in
Page 5, Line 113, intermediate, please be consistent throughout the manuscript
Page 5, Line 121, were illustrated and showed that
Page 5, Line 132, The specific distribution of different motifs may confer unique function of different genes
Page 6, Line 151, genes coordinate with annotated data
Page 7, Line 162, of which were involved genes in
Page 8, Line 195, two full stop were present in the sentence
Page 9, Line 223, table 1 cluster 1 consisted of
Page 10, Line 240, a diverse of cis-acting regulatory
Page 10, Line 247, Wounding, pathogen, and drought response cis-acting elements were
Page 11, Line 260, in some of the cotton WOX gene’s promoter
Page 12, Line 268, weighted gene co-expression network analysis (WGCNA)
Page 13, Line 271, Most cotton, most should be lower case letter
Page 13, Line 279, have the molecular function of DNA helicase activity
Page 13, Line 284-294, plant hormone, plant-pathogen interaction, homologous, protein, carbon, clustering should be lower case letter
Page 14, Line 320, For the co-expressed genes, molecular function, photosynthesis, should be lower case letter
Page 14, Line 324, Photosynthesis, should be lower case letter
Page 14, Line 341, allotetraploid species instead of specie.
Page 15, Line 363, plays roles in different stages
Page 15, Line 373, basic biological processes involved in
Page 16, Line 404, Overall, all of the cotton
Page 17, Line 493-510, the texts do not need to be bold
Page 17, Line 511, 4.6 Co-expression Betwork should be Network Analysis
Page 18, Line 513, weighted gene co-expression network analysis (WGCNA)
Reviewer 2 Report
The WOX gene family is one of the most important elements that control the development of stem cells and are involved in plant embryogenesis. The presented manuscript is a continuation of the study of the embryogenesis of cotton, an important agricultural crop. The authors of the presented manuscript received interesting and useful data.
There are some comments and questions.
100- what does "The results were almost consistent with previous studies" mean? Specify
Fig.5 What other genes besides WOX have been investigated? indicate in signature
Subsection 2.6 is better divided into two subsections: 1 - qRT-PCR for nine genes and 2 - heatmap for the all family
202 - it is not clear from Fig. 6 the distribution of genes into clusters. Specify
370 - Why was the AtWOX13 gene assigned to the WOX family if it lacks the WUS domain?
439 - significant repetition from subsection 2.8
511 - Network?
Subsection 4.5. remove bold
In references at the abbreviated of the journal a dot is put
Round 2
Reviewer 1 Report
All my concerns have been sufficiently addressed in the revision.